# Trends in the Number of Behavioural Theory-Based Healthy Eating Interventions Inclusive of Dietitians/Nutritionists in 2000–2020

**DOI:** 10.3390/nu13114161

**Published:** 2021-11-20

**Authors:** Man Luo, Margaret Allman-Farinelli

**Affiliations:** 1Charles Perkins Centre, The University of Sydney, Camperdown, NSW 2006, Australia; mluo2387@uni.sydney.edu.au; 2Sir Run Run Shaw Hospital, School of Medicine, Zhejiang University, Hangzhou 310020, China

**Keywords:** healthy eating, behaviour change theory, dietitian, nutritionist

## Abstract

Nutrition interventions developed using behaviour theory may be more effective than those without theoretical underpinnings. This study aimed to document the number of theory-based healthy eating interventions, the involvement of dietitians/nutritionists and the behaviour theories employed from 2000 to 2020. We conducted a review of publications related to healthy eating interventions that used behaviour change theories. Interventional studies published in English between 2000 and 2020 were retrieved from searching Medline, Cinahl, Embase, Psycinfo and Cochrane Central. Citation, country of origin, presence or absence of dietitian/nutritionist authors, participants, dietary behaviours, outcomes, theories and any behaviour change techniques (BCTs) stated were extracted. The publication trends on a yearly basis were recorded. A total of 266 articles were included. The number of theory-based interventions increased over the two decades. The number of studies conducted by dietitians/nutritionists increased, but since 2012, increases have been driven by other researchers. Social cognitive theory was the most used behaviour theory. Dietitians/nutritionists contributed to growth in publication of theory-based healthy eating interventions, but the proportion of researchers from other professions engaged in this field increased markedly. The reasons for this growth in publications from other professions is unknown but conjectured to result from greater prominence of dietary behaviours within the context of an obesity epidemic.

## 1. Introduction

Dietitians and nutritionists have long been translating the latest scientific knowledge into practical guidance for people to make healthy food choices. Although most people would agree that eating a healthy diet can improve health and wellbeing, few of them follow dietary guidelines. Dietitians/nutritionists are aware that many people may lack the knowledge and skills to do so and that nutrition education alone is insufficient to drive nutritional behaviour change among individuals and populations [1].

This led dietitians/nutritionists to the use of theory to inform interventions. A behaviour change theory aims to explain and understand behaviours by constructing a set of predetermined ideas that in turn allows interventionists to develop a set of strategies to elicit desirable behaviour change. Nutrition interventions developed with a sound theoretical basis may be more effective than those without a theoretical foundation [2]. Since the turn of the century, several nutrition researchers have highlighted the importance of theory-based research to test and assess successful strategies to enable individuals to change their dietary behaviour [3,4], and Contento et al. have championed theory-informed nutrition education and behaviour change for almost three decades [5].

The constructs of a behaviour change theory and the proposed mechanisms of action are underpinned with corresponding behaviour change techniques (BCTs) for intervention development. BCTs are essential not only to the design but also the analysis of an effective intervention in order to understand which BCTs work for different people in different contexts [2].

While reviews of healthy eating interventions have previously been conducted, the participation of a nutrition and dietetics professional, referred to here as a dietitian/nutritionist, in the conduct of theory-informed interventions has not been well documented. This study aimed to conduct a literature review to document the trend in the number of theory-based healthy eating interventions in the past two decades and the contributions of dietitians/nutritionists to the research field, as they are the predominant professionals trained in nutrition science and dietary counselling. Another aim was to identify the major behaviour change theories used and to document whether BCTs were described using the taxonomies [6,7].

## 2. Methods

The literature was searched in a manner consistent with systematic review best practice to find publications related to healthy eating interventions using behaviour change theories [8]. It was deemed that articles published this century, i.e., from 2000 onwards, should be included as reflective of a time commensurate with a rise in the need for healthy eating interventions as a result of the obesity epidemic and calls to action to embrace behaviour change theory for achieving much-needed dietary behaviour change [3,4]. Study eligibility criteria, search strategy and data synthesis are outlined below.

### 2.1. Study Selection Criteria

#### 2.1.1. Types of Interventions

Studies about healthy eating interventions for populations, underpinned by behaviour change theories were included. Those interventions designed without theory in the development of the intervention were excluded.

#### 2.1.2. Participants

This review included healthy populations in all age groups: infants, children, adolescents, young adults, adults and older adults. Populations with a disease, such as people with obesity, eating disorders, diabetes, cancer and cardiovascular diseases, were excluded.

#### 2.1.3. Interventions

Educational, counselling and health promotion programmes for healthier eating were included. Programmes specifically aimed at weight loss diets or medical nutrition therapy for diabetes, cardiovascular disease, coronary disease, cancer and eating disorders were excluded. The comparison condition group did not receive a healthy eating intervention.

#### 2.1.4. Outcome Variables

Outcome factors related to changes in foods or beverages consumed, dietary patterns, nutrition knowledge and other nutrition-related behaviours were included.

### 2.2. Study Designs

All interventional studies, including randomised controlled trials, controlled trials, before and after studies, and pre–post studies were included. Observational study designs were excluded.

### 2.3. Search Strategy

Search strategy was developed in consultation with an academic librarian highly experienced in searches for systematic reviews in nutrition. A concept map was developed to identify concepts and keywords for the research question. Searches were conducted in five databases: Medline, Cinahl (via EBSCO), PsycINFO, Embase and Cochrane CENTRAL. The final search was conducted on 12 April 2020. The search terms included four concepts: (1) behaviour change theories and BCTs; (2) five food groups, i.e., healthy foods and discretionary foods, i.e., unhealthy foods; (3) synonyms of “nutrition intervention”; and (4) eating behaviours, portion size and food preparation. Boolean operators were used to combine the search terms. The search was limited to human studies, English language and years 2000–2020. Appendix A shows the full search strategy in Ovid for the Medline database.

### 2.4. Screening of Search Results for Eligible Studies

All interventional studies related to healthy eating interventions using behaviour theories were included. The search results from the five databases were exported to Endnote version X8 for screening. After removing duplicates, a two-phase screening was conducted to identify relevant studies. First, the title and abstract were screened to identify papers for full-text screening and exclude those that obviously did not fit the inclusion criteria. Second, a full-text screening was performed to confirm the studies that could be included in the review and trend analysis. Only complete peer-reviewed publications were considered. Conference abstracts, editorials and dissertations were excluded. The screening was completed by one researcher (M.L.) and checked by the other (M.A.F.).

### 2.5. Data Extraction

A data extraction form was constructed based on the template in the Joanna Briggs Institute Reviewer’s Manual [9] but with some categories modified to make it more relevant to the objectives of this research.

Extracted data included general study characteristics, including publication year, country of origin and authors. The affiliation and qualification of authors were extracted to identify whether the interventions included dietitians/nutritionists. If affiliations and qualifications did not indicate author profession, a Google search of author names was conducted to discern their professional qualifications. The International Confederation of Dietetic Associations describes dietitians and nutritionists as health professionals who promote health through food and nutrition and acknowledges that the title may be interchangeable in some countries but that both titles can be recognised together and independently [10]. Data on the participant population were recorded. The dietary behaviour(s) that the interventions addressed, and the behaviour change theory used in intervention design (including BCTs if explicitly stated by taxonomy in the article) were also extracted.

### 2.6. Data Synthesis and Analysis

Determinations of the trend of the total number of publications and publications with dietitians/nutritionists involved each year from 2000 to 2020 were tabulated and plotted using Microsoft Excel 2017. To be classified as a publication inclusive of dietitians/nutritionists, only one author had to be from this professional background. Descriptive analysis of the study characteristics was compiled in tabular form. A qualitative review of the findings was synthesised.

## 3. Results

### 3.1. Study Selection

The search and screening process is presented in Figure 1 using a PRISMA flow chart. The database search resulted in 6894 articles. Of these, 3202 were duplicates, with 3692 articles remaining for title and abstract screening. The full texts of 391 articles were reviewed, and 266 articles were included for analysis and review (Appendix A shows the summary of studies).

### 3.2. Publication Trends

Figure 2 shows the trend of publications per annum about theory-based healthy eating interventions over the past 20 years. The total number of theory-based healthy eating interventions located was 261 with 266 publications [11,12,13,14,15,16,17,18,19,20,21,22,23,24,25,26,27,28,29,30,31,32,33,34,35,36,37,38,39,40,41,42,43,44,45,46,47,48,49,50,51,52,53,54,55,56,57,58,59,60,61,62,63,64,65,66,67,68,69,70,71,72,73,74,75,76,77,78,79,80,81,82,83,84,85,86,87,88,89,90,91,92,93,94,95,96,97,98,99,100,101,102,103,104,105,106,107,108,109,110,111,112,113,114,115,116,117,118,119,120,121,122,123,124,125,126,127,128,129,130,131,132,133,134,135,136,137,138,139,140,141,142,143,144,145,146,147,148,149,150,151,152,153,154,155,156,157,158,159,160,161,162,163,164,165,166,167,168,169,170,171,172,173,174,175,176,177,178,179,180,181,182,183,184,185,186,187,188,189,190,191,192,193,194,195,196,197,198,199,200,201,202,203,204,205,206,207,208,209,210,211,212,213,214,215,216,217,218,219,220,221,222,223,224,225,226,227,228,229,230,231,232,233,234,235,236,237,238,239,240,241,242,243,244,245,246,247,248,249,250,251,252,253,254,255,256,257,258,259,260,261,262,263,264,265,266,267,268,269,270,271,272,273,274,275,276]. From 2000 to2006, the number of publications was less than 11 articles in any one year [233,239,240,241,242,243,244,245,246,247,248,249,250,251,252,253,254,255,256,257,258,259,260,261,262,263,264,265,266,267,268,269,270,271,272,273,274,275,276]. Between 2007 and 2012, the number of studies per annum grew but did not exceed 15 articles [168,169,170,171,172,173,174,175,176,177,178,179,180,181,182,183,184,185,186,187,188,189,190,191,192,193,194,195,196,197,198,199,200,201,202,203,204,205,206,207,208,209,210,211,212,213,214,215,216,217,218,219,220,221,222,223,224,225,226,227,228,229,230,231,232,234,235,236,237,238]. Since 2013, the number of publications appears to have almost doubled to a maximum of 28 articles (studies) [11,12,13,14,15,16,17,18,19,20,21,22,23,24,25,26,27,28,29,30,31,32,33,34,35,36,37,38,39,40,41,42,43,44,45,46,47,48,49,50,51,52,53,54,55,56,57,58,59,60,61,62,63,64,65,66,67,68,69,70,71,72,73,74,75,76,77,78,79,80,81,82,83,84,85,86,87,88,89,90,91,92,93,94,95,96,97,98,99,100,101,102,103,104,105,106,107,108,109,110,111,112,113,114,115,116,117,118,119,120,121,122,123,124,125,126,127,128,129,130,131,132,133,134,135,136,137,138,139,140,141,142,143,144,145,146,147,148,149,150,151,152,153,154,155,156,157,158,159,160,161,162,163,164,165,166,167]. As regards the contribution of dietitians/nutritionists, they authored about 58% of publications over the two decades [13,14,15,16,18,20,22,23,24,25,27,28,30,31,33,36,37,38,39,41,44,45,48,49,54,62,63,64,66,67,72,73,75,77,80,82,85,87,92,94,95,96,98,99,101,102,107,110,114,116,119,121,122,123,124,125,126,128,132,133,135,136,138,140,144,145,146,149,151,152,153,154,156,157,158,159,160,161,162,163,164,168,169,170,171,172,174,175,176,180,184,187,189,191,192,194,195,196,198,200,201,202,203,204,206,207,209,213,214,215,216,217,219,220,221,222,224,225,226,227,230,231,232,233,236,237,239,240,243,245,246,247,248,249,250,251,252,253,255,256,258,259,260,262,263,264,265,267,268,269,272,273,274,275,276]. Before 2006, most articles published involved dietitians/nutritionists. However, the number of theory-based healthy eating interventions by researchers from other professions increased over time to make a large contribution since 2013.

### 3.3. Characteristics of the Included Studies

The publications originated from 31 different countries. The United States contributed most papers (56.4%) [12,14,15,16,19,20,23,26,27,29,33,35,37,38,39,41,43,45,49,51,55,59,62,64,66,72,73,77,78,80,83,84,85,91,95,96,98,103,107,109,110,112,115,116,119,120,121,123,124,132,134,136,138,139,140,141,142,143,146,148,151,152,153,154,155,157,158,160,161,164,168,169,170,172,174,178,179,180,181,183,184,187,190,191,195,196,198,199,200,201,202,203,204,205,206,207,208,211,212,213,214,215,217,218,219,220,222,224,225,226,227,228,230,232,235,236,237,239,240,242,243,244,245,247,248,249,250,251,252,253,254,255,256,258,259,260,261,262,263,264,265,266,267,268,269,270,272,273,275,276], followed by Iran (9.8%) [17,18,21,28,32,34,44,46,47,53,65,68,69,71,76,86,90,93,97,104,106,117,122,147,156,182], Australia (5.3%) [31,67,114,118,129,130,133,162,163,171,173,185,189,209] the United Kingdom (4.9%) [42,81,89,111,113,125,137,175,176,197,234,238,246,257], Canada (4.2%) [54,87,92,101,102,126,149,159,177,186,233,274] and small proportions from the other countries [11,13,22,24,25,30,36,40,48,50,52,56,57,58,60,61,63,70,74,75,79,82,88,94,99,100,105,108,127,128,131,135,144,145,150,165,166,167,188,192,193,210,216,221,223,229,231,241,271]. Almost one third of studies focused on children (30.8%) [15,16,22,26,27,29,31,37,38,39,43,49,50,51,55,59,62,67,69,73,75,76,77,79,81,82,84,85,88,91,92,103,110,111,114,119,120,124,131,132,133,135,138,142,144,146,147,150,153,157,158,159,160,163,164,167,169,174,175,176,179,180,181,183,187,189,194,199,200,201,205,215,216,222,226,227,232,234,237,239,242,243,245,246,247,253,256,259,263,264,267,272,273,276] and one third were inclusive of all adult age groups (29.1%) [13,14,15,18,20,21,24,31,33,35,37,39,40,43,48,49,51,52,55,56,59,65,66,70,72,73,74,75,77,78,81,83,84,85,90,94,96,101,102,103,104,105,106,110,115,120,121,126,128,132,133,134,136,139,140,143,149,161,166,167,168,172,178,184,211,212,214,217,218,220,221,229,233,235,240,250,251,252,253,255,257,260,262,265,270,271,273,274]. Adolescents and young adults accounted for 18.5% [11,17,23,25,28,32,34,40,41,47,53,54,93,95,97,100,107,109,117,118,122,123,127,141,145,151,154,155,156,158,162,171,177,188,190,191,192,197,202,204,206,207,209,210,225,232,236,241,242,244,247,248,261,266] and 14.1% [12,19,30,45,57,58,60,61,63,68,80,87,89,98,108,113,116,125,129,130,137,148,170,173,185,186,196,198,203,208,213,219,223,228,230,238,249,254,268,269,275], respectively. Few studies focused specifically on infants or older adults as the target group [36,42,44,46,64,71,86,94,99,112,152,165,182,193,195,222,224,229,231,253,258,273]. 

### 3.4. Food Behaviours Targeted by Interventions

Table 1 summarises the major dietary behaviours that theory-based nutrition interventions targeted. A totally healthy eating pattern (46.6%) [13,15,17,20,23,24,25,26,27,28,29,30,31,33,36,37,38,39,44,45,46,50,51,55,59,61,62,64,67,68,72,73,77,81,82,83,85,86,88,89,90,95,96,97,99,100,104,105,107,109,110,112,114,116,119,120,121,123,127,132,133,135,139,141,142,145,146,148,149,150,151,152,155,157,158,160,162,163,164,167,171,172,174,175,178,179,180,181,187,188,190,191,193,195,199,200,201,202,203,204,205,206,207,208,209,215,216,217,220,222,223,232,233,236,237,239,241,243,247,249,251,253,254,255,260,262,263,265,268,270,272,274] was promoted by more than half of the studies, followed by increasing fruit and/or vegetable intake (23.1%) [14,19,22,32,35,43,49,52,53,55,57,58,60,63,69,70,74,78,83,89,91,92,93,108,111,113,115,117,124,128,129,130,134,137,138,147,161,168,169,173,182,183,194,197,211,212,213,218,219,224,225,226,230,234,235,238,245,246,252,257,259,266,267,273,276], and dairy foods and calcium and/or vitamin D food source intake was also targeted (8.9%) [11,16,21,34,42,56,71,80,87,106,122,125,143,153,159,165,170,184,186,189,196,228,231,240,242]. Fewer interventions targeted dietary fat, discretionary foods and sugar-sweetened beverages (SSBs) [12,35,41,43,47,57,60,66,79,118,131,134,144,156,161,166,177,211,212,221,226,244,248,252,256,261,275]. Dietitians/nutritionists were authors in 40% or more of most publications, with the exception of ones on fruit and vegetables (38.5%) and discretionary foods.

### 3.5. Behaviour Change Theories and Techniques

Among the studies, 82.8% used one theory [11,12,14,15,16,17,18,20,21,22,24,25,26,28,29,30,31,32,34,36,37,38,39,40,41,42,43,44,45,46,47,48,51,53,54,55,56,58,60,61,63,64,65,66,67,68,69,70,71,72,75,76,77,78,79,80,81,82,83,84,85,86,88,89,90,91,92,93,94,95,96,97,98,99,101,102,103,104,105,106,108,109,110,112,113,114,115,116,117,119,120,121,122,123,124,125,126,127,128,129,130,131,132,134,135,136,137,138,139,140,141,143,145,146,147,148,150,153,154,155,156,157,158,159,160,161,162,163,164,165,166,167,168,169,170,171,172,173,174,175,176,177,178,180,181,182,184,185,186,187,189,192,193,194,195,196,197,198,199,200,201,203,204,205,206,207,209,212,213,214,215,216,217,218,219,220,221,222,223,224,225,226,227,228,229,230,232,234,235,236,238,239,241,242,243,244,245,246,247,248,249,250,254,255,258,259,260,263,264,265,266,267,268,269,270,271,272,273,274,275,276], 14.3% used two theories [19,23,27,33,35,49,50,57,59,62,73,74,87,107,111,118,133,144,149,151,152,183,188,190,191,202,210,211,231,237,240,251,253,256,257,261,262], and the remainder reported using three or more. The most common theory was the social cognitive theory (or social learning theory), accounting for almost half of the studies, followed by the theory of planned behaviour, the health belief model, the transtheoretical model and self-determination theory (see Table 2). It appeared that dietitians/nutritionists were highly represented in the studies employing social cognitive theory (65% of the total), and for the use of other theories, they comprised about half the authorship.

Figure 3 shows the percentage of total interventions using the five most popular behaviour theories in five-year intervals over the past two decades. The percentage of interventions using the social cognitive theory and the transtheoretical model decreased over time and after 2005. The theory of planned behaviour and self-determination theory grew in popularity as the theory of choice for interventions. The documentation on the use of specific BCTs using the taxonomy was poor, as only 19 studies stated the specific BCTs employed. Only six of these publications were by dietitians/nutritionists. It was noted that 6 studies published between 2011 and 2014 and 13 from 2015 onwards specified techniques.

## 4. Discussion

This study reviewed a number of publications on healthy eating interventions informed by behaviour change theory in the past two decades. From the analysis, we observed a growth in publication output, and, encouragingly, the number of interventions published with dietitian/nutritionist author input grew. This is important given that poor diet is a major contributor to the global burden of disease, and dietitians/nutritionists with expert food and nutrition knowledge should play a major role in nutritional behaviour change [278]. A marked increase in published interventions by other professions was also observed. Most publications concerned the consumption of an overall healthier diet, and this included interventions for children and for adults. Bandura’s social cognitive theory was most often reported as underpinning the intervention and has been found to be among the three most utilised theories in a previous review that identified theories of behaviour and behaviour change of potential relevance to public health interventions in behavioural and social sciences [279]. A minority of interventions specified the BCTs that they used.

The finding that the literature on theory-based healthy eating interventions has almost tripled in the past 20 years is unsurprising given the greater appreciation that changing eating behaviour is much more complex than providing individuals and the public with stand-alone evidence-based nutrition advice, as had been previously highlighted by Contento et al. decades ago [5]. Increasingly, other professions have been contributing to this field of research and publication, and one possible explanation is greater awareness of the importance of nutrition and healthy eating because of the obesity epidemic [280]. The interventions mostly targeted healthy eating in general and originated in the US. The position of the Academy of Nutrition and Dietetics [281] is that it is the total diet and overall pattern that are important for healthy eating, rather than labelling individual foods as good or bad, which is considered unhelpful. Many interventions focused on addition of healthy foods, such as fruit and vegetables, rather than giving negative messages about restriction of unhealthy foods. For fruit and vegetables, more publications were authored by groups that did not include a dietitian/nutritionist. This might be explained in the perception that the understanding of nutritional science required for such interventions may be considered less complex than for a whole-diet approach, which mostly requires multiple food targets for behaviour change and, therefore, the comprehensive nutrition knowledge that dietitians/nutritionists possess. However, while improving the population’s fruit and, in particular, vegetable intake, could appear to be easier to achieve and not require specialist nutrition knowledge, changing these behaviours is fraught with failures [282].

The most common behaviour change theories identified in the published interventions were social cognitive theory [283], the theory of planned behaviour [284], the transtheoretical model [285], the health belief model [286,287] and self-determination theory [288]. In a systematic review of 19 randomised controlled trials of theory-informed dietetic interventions in primary care, Rigby et al. [289] also found that social cognitive theory was most used. The US Academy of Nutrition and Dietetics Nutrition Care Process Terminology highlights the health belief model, transtheoretical model and social learning theory to be used in nutrition counselling with an evidence analysis published in 2010 [290]. Thus, it is unsurprising these theoretical frameworks feature most in publications by dietitians/nutritionists. The theory of planned behaviour [284] became more widely used after 2005, and self-determination theory [288] became more popular in the past decade. Both theories were first published in 1985 but appear to have taken longer to be recognised as applicable to healthy eating interventions and continue to be less often used to underpin interventions. The health action process approach (2008) emerged as being used by interventionists other than dietitians/nutritionists, but we did not see evidence of newer theories, such as the capability, opportunity, motivation, behaviour model (COM-B, 2011) and the integrated theory of health behaviour change (2009), emerging [277].

Whether theory-based interventions are more successful than those not based on a theory was outside the scope of this review, as interventions without a theory were not included. Prestwich et al. [291] reported no difference between theory-based interventions (*n* = 107) and those not based on a theory (*n* = 83).

In agreement with our study, Prestwich et al. [291] found about 90% of interventions report they are based on a theory of behaviour change, but most of them fail to clearly describe the strategies they used to change an individual’s or population’s dietary intake according to the theory. Almost none of the interventions reported links between theoretical constructs and behaviour change techniques. Our review, which included interventions published up to a decade later than that of Prestwich et al. [291], similarly found that a majority did not report on the BCTs employed, with only a minority specifying the techniques in terms of the taxonomies that Abraham and Michie [6] published of 26 behaviour change techniques in 2008, followed by a more extensive taxonomy of 93 techniques in 2013 [7]. It was observed that an increase in reporting BCTs occurred after 2010, but these interventions were mostly authored by researchers other than dietitians/nutritionists.

The authors encourage dietitians/nutritionists and other researchers to include specific behaviour change techniques employed in the description of their interventions using the taxonomy. In revealing the “active” ingredients of their intervention, it may enable us to understand why interventions succeed and fail in different populations and different contexts. BCTs can be studied as mediating factors when modelling the pathways between intervention constructs and outcomes. Perhaps the inclusion of BCTs in checklists for reporting in trials of behaviour change programmes would enable this practice.

## 5. Limitations

There are some limitations to this descriptive trend study that need to be acknowledged. Firstly, we only included published articles in English and from 2000 onwards, so this review does not encompass all original peer-reviewed studies. Publications in other languages and grey literature were not reviewed. Secondly, while we included a general term of behaviour and behaviour change technique, we also added the name of eight specific common theories as search terms, and this may have obscured finding publications with other theories. Additionally, we excluded healthy eating interventions for weight management and chronic disease, which would likely be a substantial body of manuscripts, and, therefore, this descriptive study is not exhaustive. One strength of the study is that our search strategy was designed with a librarian very experienced in nutrition searching. There may have been misclassification of papers as to whether a dietitian/nutritionist researcher was included, as this was determined from qualifications and affiliations reported in the paper supplemented by Google searching for unidentified authors.

While we have determined trends in the number of theory-based interventions, we have not sought to assess features such as most published authors and network visualisation as might be achieved with bibliometric software. We did not seek to evaluate the studies in their entirety. The effectiveness was not evaluated in depth as would be the case in a systematic review, nor did we study whether the interventions used an intervention mapping approach acknowledged as a framework for theory- and evidence-based planning in health promotion [292]. We acknowledge that dietitians/nutritionists and others involved in nutritional behaviour change take differing approaches in intervention construction and reporting of design that have not been evaluated here [293].

## 6. Conclusions

In this study, we reviewed behavioural theory-based interventions for healthy eating in the past two decades. Our analysis of 266 papers showed that publication output is increasing, and, in addition to dietitians and nutritionists, there is an increase in the number of researchers from other disciplines and multidisciplines engaged in this research field. The dominant behaviour theory used remains the social cognitive theory, but use of the transtheoretical model has lessened. However, specification of BCTs by the taxonomy remains poor. It is suggested that interventionists conducting healthy eating programmes specify the theory, the BCTs used and their links with theoretical constructs to further advance the mechanistic understanding of dietary behaviour change in nutrition and dietetics.

## Figures and Tables

**Figure 1 nutrients-13-04161-f001:**
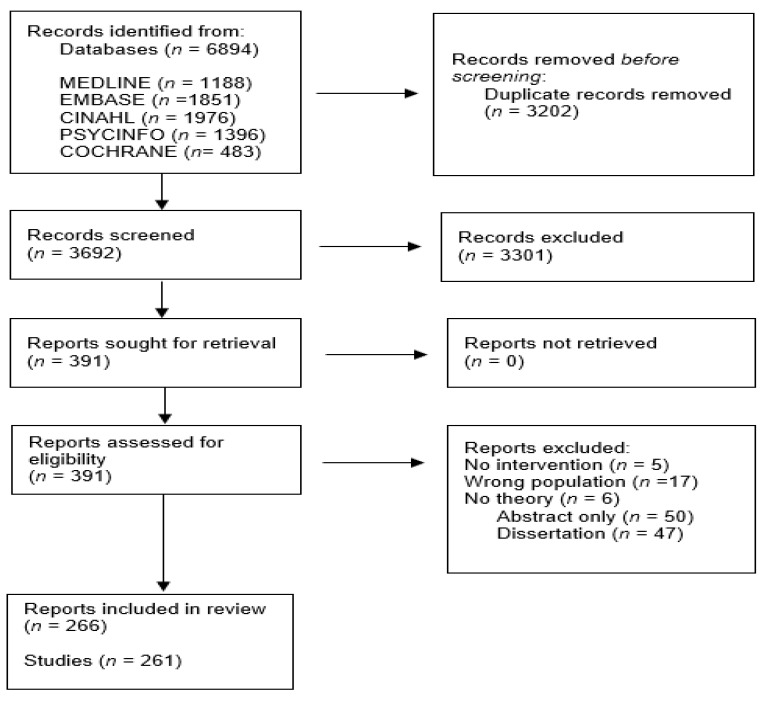
PRISMA flow chart documenting the search and study selection process.

**Figure 2 nutrients-13-04161-f002:**
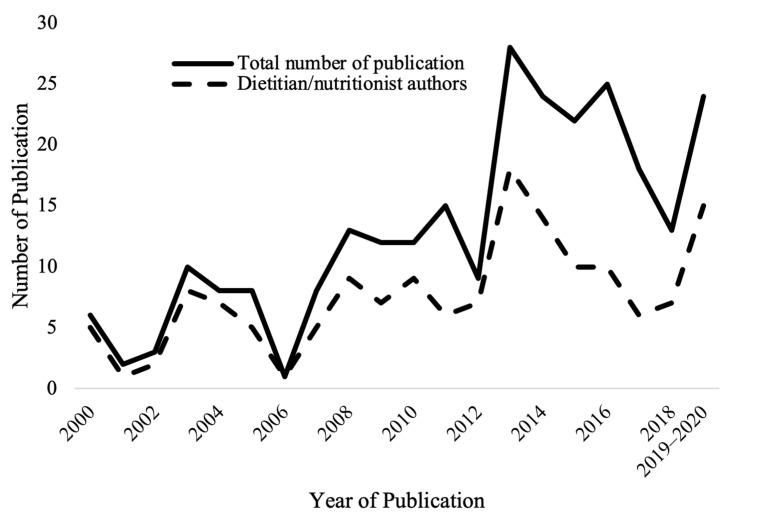
Trends in the number of publications on theory-informed healthy eating interventions between 2000 and 2020.

**Figure 3 nutrients-13-04161-f003:**
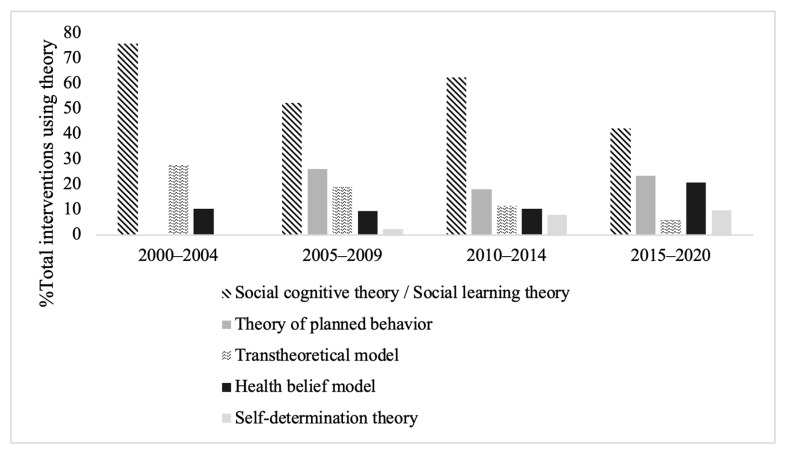
The percentage of total interventions using the top five most used behaviour change theories in five-year intervals.

**Table 1 nutrients-13-04161-t001:** List of nutrition behaviours addressed by the studies. Some interventions include more than one.

Dietary Behaviour	Number (%)	Dietitian/Nutritionist Authors *n* (%) ^a^
Global healthy eating ^b^	131 (46.6)	88 (67.2)
Increase in fruit and/or vegetable intake	65 (23.1)	25 (38.5)
Increase in dairy, calcium and/or vitamin D food source intake	25 (8.9)	13 (52.0)
Decrease in dietary fat	14 (5.0)	6 (42.8)
Decrease in discretionary food intake ^c^	6 (2.1)	2 (33.0)
Decrease in sugar-sweetened beverage intake	7 (2.5)	3 (42.9)
Others ^d^	33 (11.7)	19 (57.6)

^a^ *n* (%) is the number and percentage of total study reports with at least one dietitian/nutritionist author. ^b^ Global healthy eating refers to interventions that took a whole-diet approach to achieve increased intake of nutritious food groups and decreased intake of unhealthy foods that may be rich in deleterious nutrients, such as fat, sodium and added sugar. ^c^ Discretionary foods are those rich in deleterious nutrients, such as saturated fat, sodium and added sugar. ^d^ Others included wholegrain foods, reducing foods high in salt, decreasing snacking, increasing breakfasting, following a Mediterranean diet, increasing iron and folate intake from foods, decreasing tea or alcohol consumption, increasing water and suitable complementary/infant feeding.

**Table 2 nutrients-13-04161-t002:** Usage of different behaviour change theories in the healthy eating interventions ^a^.

Theory	Number of Studies (%)	Dietitian/Nutritionist Authors *n* (%) ^b^
Social cognitive theory/social learning theory	143 (45.8)	93 (65.0)
Theory of planned behaviour	51 (16.3)	23 (45.1)
Transtheoretical model	33 (10.6)	18 (54.5)
Health belief model	37 (11.9)	17 (45.9)
Self-determination theory	18 (5.8)	9 (50.0)
Others ^c^	30 (9.6)	19 (63.3)

^a^ A description and references for all these theories can be found in Michie et al. [277]. **^b^**
*n* (%) is the number and percentage of total study reports with at least one dietitian/nutritionist author. **^c^** Others include additional twelve theories, with more than half only used once or twice, but the following were used three to seven times: health action process approach, socioecological model of behaviour change, theory of reasoned action, and health promotion model.

## Data Availability

All included studies have their reference and data extraction available.

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
