# Peer review of "Trends in the Number of Behavioural Theory-Based Healthy Eating Interventions Inclusive of Dietitians/Nutritionists in 2000–2020"

_nutrients, 2021, doi:10.3390/nu13114161_

Round 1
Reviewer 1 Report
It is a well written paper addressing a significant topic of using theoretical frameworks in health promotion programs. However, I think that the list of studies in the paper is not an all exhaustive list of programs incorporating theoretical frameworks. I know that a few papers are not included. The authors must do another round of literature search to include all the relevant published manuscripts. Secondly, the summary of the results should be presented in a tabular format. Below please find a few more comments:
- On line 31, replace 'but' with 'and'.
- For lines 56-58, please add a reference.
- On line 109, spell out JBI.
- On line 122-124, if possible include a table for this statement.
- On line 254, please add 'also' between 'was' and 'observed'
- On line 263, publications instead of publication
- On line 271, please add a comma after vegetables
- On line 273, please omit 'of' between 'whole' and 'diet'
- On line 285, omit 'they' and add ' these theoretical frameworks'
Author Response
Thank you for giving generously of your time to review our manuscript. We apologise for our typos etc and thank you for recognising them.
We do understand your comment that we may not have exhausted every theory-based intervention. We have further highlighted this in the limitations. The search was formulated by a highly experienced librarian who has specialized skills in nutrition searching. The process was iterative with a number of revisions. We have followed recognised guidelines in our search process. Like myself you may have been disappointed to not see some of your own studies but the aim of the studies was restricted to healthy eating and we excluded interventions aimed at weight management or chronic disease which is an area that does encompass healthy eating but we would prefer to omit and perhaps consider separately. Of interest another review on healthy eating and physical activity that also included those with chronic disease (although not a direct comparison) we refer to in our manuscript (Prestwich) reported interventions from 1990 to 2008 and reported 107 theory-based and we have 293 (2000-2020).
Reviewer 2 Report
This is a review of behavioral theory=based healthy eating interventions and which have included dietitian/nutritionists. While this is important, who was included as dietitian/nutritionist was not ever defined and why this is important is never made clear. Additionally the secondary aim of inclusion of BCTs seems to creep in as more important. Yet, it is not justified why BCTs should be looked at more than more comprehensive behavioral intervention designs such as Intervention Mapping and Nutrition Education DESIGN Procedure. These need to be addressed along with my other comments below for this manuscript to be recommended for publication.
Lines 38-41. Isobel Contento has been one of the leading voices for theory-based and behaviorally focused nutrition education. Here comprehensive review: 26. Contento, IR, Senior author. [add other authors The effectiveness of nutrition education and implications for nutrition education policy, programs and research. A review of research. J Nutr Educ. 17:279-418, 1995.
27. and
Contento IR, Koch PA. Nutrition Education: Linking Research Theory and Practice. Fourth Edition. Burlington MA: Jones & Bartlett Learning. 2021.
Line 47-49: Need some justification for why it is important to document the participation of dietetics professionals (what do they add that makes it important to know if, how often, and in what ways they are involved. Note: I am not saying this is not important (and I am an RD). However without a justification there is no basis for this paper.
Line 51: Here says dietitians/nutritionists when above says dietetics professional. Please make these consistent.
Also either here or in line 123 you need to define dietetics professional (probably this is having the RD/RDN credential). You also need to define nutritionist. There is no standard or legal definition of nutritionist, anyone can say they are a nutritionist so you must have had some definition and you need to clearly define this so the reader knows who was included and who was not. Theoretically, any article had people who worked in nutrition so they were all nutritionist. More specifically, how did you define what training in nutrition science or dietary counseling? Without this the reader has no idea which authors you would have included or excluded. How did you look up what degrees authors had to know this? As another thought did degrees in nutrition education or behavior nutrition were included or only nutrition science. Also, for a paper with many authors was it enough to have one author that met your definition, no matter where that author was on the list of authors? This needs much more clarity it is glossed over in this manuscript and is your key outcome.
Line 147 says involves dietitians/nutritionist. Involves is a very vague term please define more clearly.
Line 67 need period
Section 2.1.2 your “unit” was published article you need to make it clear that these articles evaluated interventions with healthy populations.
Section 2.1.3 First sentence is what was excluded. You need more on what were the types of interventions included, something such as intervention aimed at changing dietary behaviors (e.g., eating more fruits and vegetables, drinking fewer sweetened beverages, eating fewer ultra-processed foods) that could promote health and included behavior change theories and/or BCTs. The second sentence seems like it goes more with participants (about comparison groups).
Figure 2: This adds more confusion as this says “dietitians” above you say nutrition scientists. Again, this needs much more clearly.
Line 186: “dietary factors” is typically defined as the “behavior change goal” which makes it clearer that theory is being used to change behavior not just provide information about dietary factors.
Table 1:
1) This says “nutrition targets” then in the table the header is “dietary factors” Again suggest changing this to behavior change goal.
2) This says dietitian/nutritionist authors as above please be clearer on this and this should match figure 1 which only says dietitian.
3) Column 1 all need a direction — these intervention aim to increase or decrease consumption of these foods also for the nutrients (calcium, vitamin D, dietary fats) it is better to say foods with these as that is what is discussed in education (if it is done well). Also “global health eating” needs a footnote to described what were the behaviors discussed in these interventions — theory-based interventions do not work if they just tell people “globally eat healthfully.”
Line 208: Above you say that stating a behavior change theory was an inclusion criteria, justify why these three were included, if you included interventions where a behavior change theory was not stated I believe your n would have been much larger.
Line 219 again here saying dietitian/nutritionist
Line 239-241. What evidence do you have that BCTs were used as a result of other health researchers such as psychologists being involved. Additionally this is the only time what other types of professionals are involved in this research is included.
Design and results general: Many have proposed that how interventions are designed (that is more specifics on how the the psycho-social determinants are used) is also crucial for intervention, which is different, and perhaps more comprehensive than the application of BCTs — which is how the determinants are activated. Intervention mapping (Bartholomew-Eldredge et al ) and Nutrition Education DESIGN Procedure (Contento and Koch) are two examples. This are broader as they go beyond how determinants are applied (BCTs) to include how interventions are sequenced and how they are delivered communication). You need to justify why looking at these were not included in this study.
Line 253-254: You have to state why this is a concern, or is it a concern to you? Since you did not define why it is (or is not) important for dietitians/nutritionists being involved above, all you do here is restate the result. This needs more discussion. This is the key aim of your study.
Line 254 Sentence that starts most publications… is not clear. Next sentence is a fragment. Also once this is clear, you need to discuss more on if you think it is it good or bad to be so general in the dietary intent of the intervention. Those who work in behavioral nutrition would say that is an extremely disturbing finding, that half of the intervention were “global healthy eating.” Again, behavior change theory-based interventions have been found to be effective only when specific behaviors are discussed so that people know what to change. “Eating healthy” is not specific.
Line 297 again explain why three with no theory included.
Line 303 how does Prestwich define strategies and does this include Intervention Mapping and/or Nutrition Education DESIGN procedure?
Line 313 Dietitians and other researchers — this paragraph makes it seem like this paper is really about BCTs not about nutrition professionals which makes the purpose of this paper less clear, as discussed above.
Conclusion: Edit once all other edits are made once again, your aim was about dietitians/nutritionists yet this is neutral on why this was your main study aim and again makes it seem like this was really about BCTs.
Author Response
A sincere thank you to the reviewer for giving so generously of their time and highlighting other important influential theory-based nutrition education. We understand that reviewing in such detail takes much effort. Every point has been carefully considered and either lead to changes in the manuscript or provision of further explanation for the reviewer. We very much appreciate their experienced views on the design of nutrition education and dietary change. Please understand the journal practice offers limited revision time. The changes are extensive and in track changes
As the reviewers comments are extensive we will upload as a separate file.

Round 2
Reviewer 1 Report
You did not address my comment about "Secondly, the summary of the results should be presented in a tabular format."
Please address that and I can review the manuscript again. Thanks.
Author Response
We sincerely apologise we did not appear to address your request for a table. We apologise have missed your point. We inserted the new Table 1 which addressed the main aim of the study to count the frequency of studies on healthy eating using a theory and those identified as conducted by dietitian/nutritionists similar to a bibliometric analysis. We are not certain what is required. We do have the complete citation of every study used in the reference list. Can we have clarification of what is required please.
Reviewer 2 Report
Overall this manuscript has been improved tremendously by addressing the comments of the reviewers and it now makes a valuable contribution to the literature.